# Novel Sulfonylurea Derivatives as Potential Antimicrobial Agents: Chemical Synthesis, Biological Evaluation, and Computational Study

**DOI:** 10.3390/antibiotics12020323

**Published:** 2023-02-03

**Authors:** Fan-Fei Meng, Ming-Hao Shang, Wei Wei, Zhen-Wu Yu, Jun-Lian Liu, Zheng-Ming Li, Zhong-Wen Wang, Jian-Guo Wang, Huan-Qin Dai

**Affiliations:** 1State-Key Laboratory of Research Institute of Elemento-Organic Chemistry, Frontiers Science Center for New Organic Matter, College of Chemistry, Nankai University, Tianjin 300071, China; 2Scientific Research Training Center for Chinese Astronauts, Beijing 100094, China; 3State Key Laboratory of Mycology, Institute of Microbiology, Chinese Academy of Sciences, Beijing 100101, China; 4Savaid Medical School, University of Chinese Academy of Sciences, Beijing 100049, China

**Keywords:** MRSA, sulfonylurea derivative, antimicrobial agents, DFT calculation, structure–activity relationships

## Abstract

Methicillin-resistant *Staphylococcus aureus* (MRSA) is a worldwide health threat and has already tormented humanity during its long history, creating an urgent need for the development of new classes of antibacterial agents. In this study, twenty-one novel sulfonylurea derivatives containing phenyl-5-vinyl and pyrimidinyl-4-aryl moieties were designed and synthesized, among which, nine compounds exhibited inhibitory potencies against Gram-positive bacterial strains: MRSA (Chaoyang clinical isolates), *S. aureus* ATCC6538, vancomycin-resistant *Enterococci*-309 (VRE-309), and *Bacillus subtilis* ATCC 6633. Especially, **9i** and **9q** demonstrated inhibitory activities against the four bacterial strains with minimum inhibitory concentrations (MICs) of 0.78–1.56 μg/mL, and quite a few of other MRSA clinical strains with MICs of 0.78 μg/mL, superior to those of the positive controls vancomycin (MIC of 1 μg/mL) and methicillin (MIC of >200 μg/mL). This is the very first time that sulfonylurea derivatives have been identified as promising inhibitors against different MRSA clinical isolates. In addition, all the MIC values of the synthesized compounds against *Candida albicans* were greater than 100 μg/mL. Since the reported anti-*Candida* activities of sulfonylureas were due to acetohydroxyacid synthase (AHAS) inhibition, the molecular target against MRSA for the target sulfonylureas was thought to be a different mode of action. Density functional theory (DFT) calculations were finally performed to understand the structure–activity relationships, based on which, significant differences were observed between their HOMO maps for compounds with strong antibacterial activities and weak anti-MRSA effects. The present results hence provide valuable guidance for the discovery of novel agents to treat bacterial infections, especially against MRSA.

## 1. Introduction

*Staphylococcus aureus* is one of the main causes of hospital and community infections, which are distributed all over the globe. With the increasing drug resistance to antibiotics, the treatment and control of *S. aureus* infection is becoming more and more difficult [1]. Since the first emergence of methicillin-resistant *Staphylococcus aureus* (MRSA) in the UK in 1961, it has spread rapidly and shown multiple drug resistance worldwide within recent years [2,3], including to *β*-lactams, quinolones, aminoglycosides, glycopeptides, and macrolides [4,5]. Nowadays, MRSA infections have exceeded AIDS, tuberculosis, and viral hepatitis, threatening human public health seriously [6]. Vancomycin has remained one of the first-line clinical drugs since 1958 [7]. However, the minimum inhibitory concentration (MIC) against MRSA has shifted upward due to its overuse [8,9,10]. In 1988 the first case of vancomycin-resistant *Enterococci* (VRE) was identified [11], and the first isolate of vancomycin-resistant *S. aureus* (VRSA) was also reported in the USA in 2002 [12,13,14]. Since then, at least 14 VRSA strains in total have been isolated. Therefore, there exists a significant need to discover alternative antibacterial agents to defeat the increasing bacterial resistance.

The sulfonylurea herbicides were developed in the 1970s, among which chlorsulfuron was the first commercial product. The molecular structures of sulfonylureas are simple and the biological target for the herbicidal activity is acetohydroxyacid synthase (AHAS, EC 2.2.1.6), which exists only in plants and microbes, and therefore, this family of herbicides is safe towards mammalian bodies [15]. In addition, some other sulfonylureas, such as the licensed glipizide, have been well-known for several decades as antidiabetic drugs targeting pancreatic β-cells [16]. Moreover, some antimicrobial sulfonamides such as sulfamethazine were approved many years ago and combine the structural features of sulfonylurea and *p*-aminobenzoic acid [17]. It is only recently that a few sulfonylureas were reported to display promising inhibitory activities against the human pathogens *Candida albicans* and *Mycobacterium tuberculosis* [18,19,20,21] due to AHAS inhibition. However, to date, there are no published reports of sulfonylurea derivatives as desirable inhibitors of MRSA.

Previously, Wei et al. found that some sulfonylureas containing aryl-5-alkenyl moieties displayed considerable activities against plant fungi strains [22]. Especially, compound **A** (Figure 1) exhibited in vitro inhibitions against *Ceratobasidium cornigerum* at a level comparable with that of chlorothalonil with EC_50_ values of 4.54 μg/mL and 4.45 μg/mL, respectively. When the bromide atom in compound **A** was changed to chlorine to create a new compound **B**, it showed much stronger in vitro inhibitions (MIC of <0.05 μg/mL) against *C. albicans* SC5314 than the antifungal drug fluconazole (MIC of 1.56 μg/mL) [23]. Nevertheless, Compound **B** possessed very weak bioactivity (MIC of >40 μg/mL) against *S. aureus* ATCC6538 [23]. Given that the anti-*Candida* activities of compound **B** and other sulfonylurea compounds are caused by AHAS inhibition [18,19,20,23], it suggests that AHAS might not be an ideal target to inhibit *S. aureus*. The selective inhibitions against *C. albicans* strains of these compounds were achieved herein. In an effort to discover novel antibiotics to combat bacterial resistance, twenty-one target compounds containing phenyl-5-vinyl and pyrimidinyl-4-aryl moieties were designed and synthesized (Figure 2). The biological activities against Gram-positive bacterial species (MRSA, *S. aureus*, VRE, and *B. subtilis*) and *C. albicans* were determined for the discovery of novel antimicrobial agents. At the same time, for comparison, we evaluated the antibacterial effects of ten commercial sulfonylurea herbicides in the present research. In addition, the in silico quantum chemistry of selected compounds was also investigated to elucidate the structure–activity relationships. The chemical structures of chlorsulfuron, sulfamethazine, glipizide, compound **A**, and compound **B** are shown in Figure 1.

## 2. Results and Discussion

### 2.1. Chemistry of the Compounds

As illustrated in Figure 1, intermediate **3** could be synthesized via the condensation of substituted aryl-methyl-ketone and ester when NaH was selected as the base to promote the complete conversion of substituted aryl-methyl-ketone to enol anion and to improve the yield. Compound **3** was purified by vacuum distillation to achieve a higher purity. Intermediates **4a–i, 4m–p,** and **5a–d** were obtained through a dehydration cyclization of **3** or **2** with guanidine hydrochloride in dry EtOH to give a high yield, for which EtONa was used as the base. **4j–l** were prepared by the bromination of **4b**, **4g**, and **4h** in a mixture of CH_3_CN and NBS. In addition, compounds **4q** and **6a-c** had to be synthesized via a chlorine substitution of compounds **5a–d** in phosphorus oxychloride solvent, and ClSO_3_H was chosen as the catalyst. Subsequently, compounds **4r-u** were obtained through a nucleophilic substitution by mixing the chlorine substitution products **4q** or **6a–c** with MeONa under reflux in pre-dried MeOH. The sulfonyl carbamate **8** was produced via a nucleophilic substitution by mixing sulfonamide **7**, ethyl chloroformate, and K_2_CO_3_ in refluxing acetone. Care should be taken when handling ethyl chloroformate as it is highly toxic.

**9a–u** were obtained through the substitution of sulfonyl carbamate **8** by heterocyclic amines **4a–u** with continuous removal of the azeotrope mixture of toluene and ethanol; after that, the samples were further purified by column chromatography. The target compounds were characterized by ^1^H NMR, ^13^C NMR, and HRMS. Table 1 lists the variations of the different substituents of the sulfonylureas.

In order to further confirm their chemical structures, a single crystal of **9j** (CCDC number: 2168485) suitable for X-ray diffraction was successfully obtained using a self-evaporation method in acetone solution, as shown in Figure 2. The bond angles of COOC–SO1–OOO4, OOO4–SO1–NOOA, and NOOA–SO1–OOO6 were 111.43°, 104.57°, and 109.57°, respectively, indicating the sp^3^ hybridization state of the SO1 atom. The sum angle of OOO5–COOI–NOOA (122.14°), OOO5–COOI–NOO9 (121.48°), and NOOA–COOI–NOO9 (116.38°) was 360°, suggesting a plane sp^2^ hybridization state of COOI. The torsion angle between the benzene ring and the plane (COOG, COOQ, COOR) was 6.98°, showing that the benzene ring and the vinyl were not in a same plane. The dihedral angle between pyrimidine and thiophene was 4.96°, implying that these two rings were not coplanar either. The dihedral angle between the benzene ring and pyrimidine was 80.17°.

### 2.2. The In Vitro Antimicrobial Activities and Structure–Activity Relationship (SAR)

The preliminary antibacterial and antifungal activities of the synthesized sulfonylurea compounds are presented in Table 1. MRSA(Chaoyang clinical isolates), *S. aureus* (ATCC 6538), VRE-309, and *B. subtilis* (ATCC 6633) were selected as the Gram-positive bacteria strains, and *C. albicans* (SC 5314) was chosen as the fungus strain. We also evaluated the antibacterial activities of ten commercial AHAS-inhibiting sulfonylurea herbicides (chlorsulfuron, metsulfuron methyl, ethametsulfuron, chlorimuron ethyl, tribenuron methyl, sulfometuron methyl, thifensulfuron methyl, pyrazosulfuron ethyl, bensulfuron methyl, and ethoxysulfuron, as drawn in Appendix A) for comparison. Considering the biological results, all the MIC values were >100 μg/mL for the commercial sulfonylureas, revealing that these compounds did not have inhibitory effects on the Gram-positive bacteria. However, among the ten compounds, chlorimuron ethyl and ethoxysulfuron were reported to be very strong inhibitors against *C. albicans* with MIC_50_ values of 2 μM for each of them [19]. Excitingly, considering the synthesized compounds in this study, several demonstrated potent activities upon the tested bacteria strains. Among them, **9i** and **9q** exhibited the highest biological activities, with MIC values of 0.78–1.56 μg/mL for MRSA (Chaoyang clinical isolates), *S. aureus* (ATCC 6538), VRE-309, and *B. subtilis* (ATCC 6633), respectively. The inhibitory effects of **9i** and **9q** against MRSA (Chaoyang clinical isolates) and *S. aureus* (ATCC 6538) were stronger than those of vancomycin and methicillin. It should be noted that for vancomycin, the MIC value was >16 μg/mL against VRE-309, while the corresponding data for these two compounds were both 1.56 μg/mL. Besides **9i** and **9q**, compounds **9a**, **9b**, **9d**, **9e**, **9h**, **9o**, **9r**, and **9t** also exhibited strong to moderate inhibitory potencies against the tested bacteria with MIC values ranging from 6.25 μg/mL to 100 μg/mL. For the rest of the compounds, all the MIC values against the Gram-positive strains were over 100 μg/mL. On the other hand, the anti-*Candida* susceptibilities for all the synthesized sulfonylurea compounds were all >100 μg/mL, showing that they were not ideal inhibitors for antifungal research. In our previous research [23], a series of 2,5–substituted sulfonylureas were chemically synthesized and biologically evaluated. It was interesting that some of those compounds showed strong antifungal activities against *C. albicans* (SC 5314); however, none of them showed any effects against *S. aureus* (ATCC 6538) at 40 μg/mL concentration. In contrast, a few of the current sulfonylurea compounds were very active towards *S. aureus* and no anti-*Candida* activities were observed at the highest concentration. Taken together the above findings show it is quite likely that the anti-MRSA efficiencies of the present sulfonylureas were not a result of AHAS inhibition. The exact mode of action will have to be clarified by further extensive screening of possible biological targets for antibacterial agents. A good case in point is N5-carboxy-amino-imidazole ribonucleotide (N5-CAIR) mutase (PurE, EC5.4.99.18), a vital enzyme involved in the de novo purine biosynthesis pathway in organisms, which has been validated as a successful target for the design of anti-MRSA agents [24,25]. In further research, related biochemical assays will be carried out to determine whether PurE is a possible target for the anti-MRSA sulfonylureas.

To validate whether the active compounds possess a broad antibacterial spectrum, the anti-MRSA activities of **9i**, **9q**, **9b,** and **9e** were further explored against a series of clinically isolated MRSA strains (309-4, 6281, 309-8, 6-42, 8-21, 309-3, 309-1, 309-7, 8-24, and 309-6) following a protocol similar to that used previously [26]. Excitingly, the compounds identified with potent activities in Table 1 were also very strong inhibitors against the other ten MRSA strains, as summarized in Table 2. The MICs of **9i** and **9q** were 0.78–1.56 μg/mL against the various MRSA strains, while the corresponding data for **9b** and **9e** were 6.25–12.5 μg/mL at these conditions. The biological results for the four selected compounds against the ten MRSA isolates showed general agreement with the MIC values against MRSA (Chaoyang clinical isolates) and *S. aureus* (ATCC 6538), indicating that these compounds were likely to display similar efficiencies towards other MRSA strains which were absent in our experiments. The control drugs vancomycin and methicillin had MIC values of 1.0 μg/mL and >200 μg/mL, respectively, against all the MRSA strains. To our knowledge, the sulfonylurea compounds have never been reported to be effective agents against any strains of Gram-positive *S. aureus*, let alone the MRSA isolates. Therefore, the anti-MRSA activities of the compounds herein will shed some bright light on the design and discovery of new anti-MRSA inhibitors with distinct molecular skeletons.

In order to analyze the structure–activity relationship, a comparison was made between the target compounds and the previous synthesized compounds [23], since there existed obvious difference in the biological behaviors of *S. aureus* and *C. albicans*, as mentioned above. The substituted groups in the heterocycle ring of the compounds published in that article were –SCH_3_, -CH_3_ and –OCH_3_, while in the present research, an aromatic ring or substituted aromatic ring was attached to the pyrimidine ring. Hence the selective inhibitions against bacteria or fungus of the sulfonylurea derivatives could be achieved by changing the substituents in the pyrimidine ring. For the target compounds here, although the major variable was the R^3^ group, R^1^ and R^2^ also played important roles affecting the MRSA (Chaohyang) inhibition. For example, the only difference among **9d**, **9i**, and **9l** were the R^1^ groups (-H, -CH_3_, and -OCH_3_); however, **9i** had the lowest MIC, and **9l** did not show any activity in the assay. Another sound instance was the group of **9j**, **9m**, and **9q**, the R^1^ groups for which were -CH_3_, -OCH_3_, and -H, and it was interesting that the MICs for them against MRSA were >100 μg/mL, >100 μg/mL, and 0.78 μg/mL, respectively. Although a few sulfonylurea derivatives have demonstrated considerable activity against Gram-positive bacteria strains, especially MRSA from a variety of clinical sources, it is difficult for us to formulate a simple and general trend to explain the structure–activity relationships. The next round of molecular design is essential to synthesize dozens of new compounds and a comprehensive biological evaluation will discover more potent anti-MRSA inhibitors, based on which, a more reliable and detailed discussion will be provided for the understanding of the structural feature or pharmacophore of the active compounds. With no doubt, **9i** and **9q** are perfect lead compounds to design the second generation of anti-MRSA sulfonylureas.

### 2.3. Quantum Calculation Analysis

Frontier molecular orbital (FMO) is usually considered to play an important role in the biological activities of medicinal or agrochemical agents [27,28], and several quantum studies have been undertaken for the sulfonylurea compounds using density functional theory (DFT) calculation to reasonably explain the difference in herbicidal activities or AHAS inhibitions of compounds with different chemical structures [23,29,30,31]. Keeping this idea in mind, the best anti-MRSA compounds **9i** and **9q** were chosen for the DFT chemistry, and two other compounds **9u** and **9j** were also subjected to quantum calculation. This was for two reasons: (1) **9u** and **9j** did not exhibit any antibacterial activities, even at the highest concentration of 100 μg/mL; and (2) **9u** had a similar structure to **9i**, whereas the structure of **9j** was close to that of **9q**. From Figure 3, it can be seen that the lowest unoccupied molecular orbital (LUMO) maps for all the investigated compounds are fairly similar, which mainly cover the heterocycle ring and the substituents attached to this moiety. Interestingly, differences can be observed in the highest occupied molecular orbital (HOMO) maps between the potent anti-MRSA inhibitors and the compounds inactive against MRSA. For **9i** and **9q**, the HOMO maps are located mainly on the phenyl ring connected with the –SO_2_- group; however, for **9u** and **9j,** although the majority of HOMO maps are distributed on the phenyl ring, an obvious distribution can also be found on the heterocycle ring. The difference in HOMO maps is likely to be a possible factor accounting for the anti-MRSA effects, which should be considered for further design of new antibiotics belonging to this family to combat resistant Gram-positive bacteria.

## 3. Materials and Methods

### 3.1. Regents and Instruments

The general synthetic procedure for title compounds and intermediates is described in Figure 1. The starting materials were purchased from the following commercial suppliers: Alfa-Aesar, Sigma-Aldrich, TCI, Apichemical and Chemieliva, J&K Chemical, Accela ChemBio etc., which were all analytical grade in purity. All solvents and liquid reagents were pre-dried using standard methods and distilled before use. The main deuterated reagent was dimethyl sulfoxide (DMSO-*d*6). For the structural and spectroscopic characterization of target compounds, an X-4 binocular microscope melting point apparatus (Beijing Tech Instrument Co., Beijing, China, uncorrected), a 400 MHz Bruker AV 400 nuclear magnetic resonance spectrometer (NMR) (Bruker Co., Fällanden, Switzerland), an Agilent 6520 Q-TOF LC/MS high-resolution mass spectrometer (HRMS) (Agilent Co., Santa Clara, CA, USA), and a Rigaku Saturn 70 CCD diffractometer (Rigaku Corporation, Tokyo, Japan) were used.

### 3.2. Synthesis of Intermediates **2–5**

#### 3.2.1. Synthesis of Intermediates **2** [32]

*N*,*N*-dimethylformamide dimethyl acetal (0.15 mol) was added to a solution of compound **1** (0.1 mol) in 80 mL of toluene, and then the solution was heated to 110 °C. The low boiling point azeotrope mixture of toluene and methanol was evaporated every 4 h. The methanol generated in the reaction mixture was removed continuously to accelerate the reaction rate for the preparation of intermediate **2.** The reaction system was monitored by TLC every 2 h until its completion at 8 h. It was allowed to cool to room temperature and 400 mL n-hexane was added under stirring conditions. A large amount of white solid formed, and the crude product **2** was obtained after suction filtration and being air-dried, and this was consumed directly in the next step without further purification.

#### 3.2.2. Synthesis of Intermediates **3** [33]

To a suspended mixture of NaH (0.28 mol) in 500 mL of dry toluene was added ethyl acetate (or diethyl carbonate) (0.2 mol), and the mixture was heated to 110 °C. After that, a solution of compound **1** (0.1 mol) in 150 mL toluene was added dropwise, and the H_2_ gas was released from the mixture at the same time. The system was allowed to cool to ambient temperature, and 30 mL of acetic acid was added dropwise when the H2 gas was not generated in the reaction solution. With the addition of acetic acid, viscous solids appeared, and most of them were dissolved in toluene after dropping. 500 mL of water was added to the reaction mixture under stirring conditions, after which the organic phase was collected and dried with anhydrous Mg_2_SO_4_. The organic solution was subsequently evaporated under reduced pressure and the residue was distilled under vacuum to obtain intermediates **3**.

#### 3.2.3. Synthesis of Intermediates **4a–i** and **4m–p** [34]

Guanidine hydrochloride (10 mmol) and intermediate **2** (10 mmol) were added to a solution of Na (11 mmol) in 30 mL of dry ethanol. After that, the solution was heated to 78 °C and maintained at this temperature for 10 h. Subsequently, the reaction mixture was cooled to 25 °C. After removal of the solvent, the residue was poured into 200 mL of ice-water under vigorous stirring conditions, and a large amount of solid appeared. The precipitate was filtrated and dried in vacuo to give **4a–i** without further purification.

**4m–p** were obtained via a similar synthetic procedure.

#### 3.2.4. Synthesis of Intermediates **4j–l** [35]

A solution of NBS (*N*-bromosuccinimide) (6 mmol) in 10 mL of CH_3_CN was added dropwise into the solution of intermediate **4g** (5 mmol) in 10 mL of CH_3_CN at ambient temperature. After that, the reaction mixture was slowly heated to 75 °C and monitored by TLC every 1 h until it was complete at 3 h. The solvent was then evaporated under reduced pressure, and the residue was dissolved in 30 mL of ethyl acetate. The organic solvent was washed with 30 mL of saturated aqueous NaHCO_3_ solution, and dried with anhydrous Mg_2_SO_4_ in the next step. After removal of the ethyl acetate, the crude product was purified by column chromatography on silica gel eluting with petroleum and petroleum/ethyl acetate (5:1 gradient elution, *v*/*v*) to obtain **4j**.

**4k** and **4l** were prepared via a similar synthetic procedure.

#### 3.2.5. Synthesis of Intermediates **5a–d**

The synthesis of intermediates **5a–d** was similar to that of intermediates **4m–p**. The pH value of the solution of residue in water was adjusted to 7 with concentrated hydrochloric acid, and then the solid formed.

#### 3.2.6. Synthesis of Intermediates **4q** and **6a–c** [36]

Intermediate **5a** (2-amino-6-phenylpyrimidin-4-ol,5 mmol) and chlorosulfonic acid (1 mmol) were added to phosphorus oxychloride (50 mmol). After that, the reaction was slowly heated to 45 °C, maintained at that temperature for 0.5 h, and then heated to 90 °C. After an additional 4 h, the reaction mixture was transparent and cooled to ambient temperature. To the reaction mixture, 200 g of crushed ice was added, and the pH value of the solution was adjusted to 7–8 using 25% ammonium hydroxide or aqueous NaOH solution. The solution was extracted 3 times with ethyl acetate (50 mL, 30 mL, and 20 mL, respectively). The combined organic phase was washed with saturated aqueous NaHCO_3_ solution, and dried with anhydrous Na_2_SO_4_. The solvent was removed under reduced pressure, and the residue was finally purified by column chromatography on silica gel eluting with petroleum ether and petroleum ether/ethyl acetate (1:1 gradient elution, *v*/*v*) to obtain **4q**.

**6a–c** were prepared via a similar synthetic procedure.

#### 3.2.7. Synthesis of Intermediates **4r–u**

Taking 4-methoxy-6-phenylpyrimidin-2-amine (**4r**) as an example: 4-chloro-6-phenylpyrimidin-2-amine (**4q**) (5 mmol) was added to a solution of Na (10 mmol) in 30 mL of dry methanol, and then the reaction mixture was heated to 65 °C and monitored by TLC every 2 h till it finished at 4 h. The solvent was evaporated in a vacuum, and the crude product was purified by column chromatography on silica gel eluting with petroleum ether and petroleum ether/ethyl acetate (1:1 gradient elution, *v*/*v*) to obtain **4r**.

**4s–u** were prepared via a similar synthetic procedure.

### 3.3. Synthesis of Target Compounds **9a–u** [37]

Potassium carbonate (4 mmol) and ethyl chloroformate (2.2 mmol) were added to a solution of 2-chloro-5-vinylbenzenesulfonamide (**7**) (2 mmol) in acetone. The reaction mixture was heated to 56 °C and monitored by TLC every 1 h until its completion at 4 h. To the suspended mixture, 10 mL of aqueous hydrochloric acid (1 M) solution was added to quench the reaction. The organic solvent was evaporated under vacuum and the residue was extracted 3 times with ethyl acetate (50 mL, 30 mL, and 20 mL, respectively). After that, the combined ethyl acetate was washed with 30 mL of saturated aqueous NaHCO_3_ solution and dried with anhydrous Mg_2_SO_4_. The organic solvent was evaporated under reduced pressure, and the residue (**8**, ethyl ((2-chloro-5-vinylphenyl)sulfonyl)carbamate) was dissolved in 40 mL of toluene before intermediates **4a–u** (2 mmol) were added to the solution. The reaction mixture was slowly heated to 110 °C and monitored by TLC every 3 h till it completed at 9 h. The solvent was evaporated in vacuum when it was cooled to ambient temperature. The residue was purified by column chromatography on silica gel eluting with dichloromethane and dichloromethane/methanol (50:1 gradient elution, *v*/*v*) to obtain **9a–u**.

The synthetic procedure of 2-chloro-5-vinylbenzenesulfonamide (**7**) was detailed in our previous work [23].

The full characterization data of the intermediates and target compounds are provided in the Appendix A.

### 3.4. Bacterial Strains and In Vitro Antibacterial Susceptibility and MIC Determination

The Gram-positive bacterial species and strains used in this research were MRSA clinical isolates (Chaoyang, 309-4, 6281, 309-8, 6-42, 8-21, 309-3, 309-1, 309-7, 8-24 and 309-6), *S. aureus* (ATCC 6538), VRE (VRE-309), and *B. subtilis* (ATCC 6633). MRSA (Chaoyang clinical isolates) is a clinical isolated strain from Chaoyang Hospital in Beijing. MRSA (309-4, 309-8, 309-3, 309-1, 309-7 and 309-6) and VRE-309 are clinical isolated strains from the 309th hospital of the Chinese People’s Liberation Army in Beijing. MRSA (6281, 6-42, 8-21 and 8-24) are clinical isolated strains from the 306th hospital of the Chinese People’s Liberation Army in Beijing. *S. aureus* (ATCC 6538) and *B. subtilis* (ATCC 6633) were from Institute of Microbiology, Chinese Academy of Sciences.

The procedure for the antimicrobial bioassay was implemented according to the method reported in the literature [31,38]. Twenty-one target compounds were prepared in 4 mg/mL sterile dimethyl sulfoxide (DMSO) stock solution, and the stock solution was serially diluted by sterile DMSO. The bacterial strains were stored as glycerol stocks in −80 °C conditions and were streaked onto Mueller–Hinton agar (MHA) for colony growth at 37 °C before use. The inhibitory activities and MICs were detected in flat bottom, 96-well microtiter referring to a broth microdilution protocol modified from the Clinical and Laboratory Standards Institute (CLSI) M7-A6, M-38A, and M-27A2 methods. Subsequently, single colonies of bacteria were picked from MHA plate and adjusted to approximately 10^5^ CFU/mL using Mueller–Hinton Broth (MHB) as a bacterial suspension. To each row on 96-well plates containing 78 mL of bacteria suspension in each well were added aliquots 2 mL of 2-fold serial dilution of every compound (in DMSO). The 96-well plates were incubated aerobically at 37 °C for 16 h before inhibitory results were recorded. MICs were defined as the minimum inhibitory concentrations of the test compounds that can inhibit visible bacterial growth after 16 h incubation. The MICs were measured twice in triplicate. For the bioassays, vancomycin and methicillin were selected as control drugs for the MRSA clinical isolates *S. aureus* (ATCC 6538), VER-309 and *B. subtilis* (ATCC 6633).

### 3.5. Human Pathogen Fungus and In Vitro Antifungal Susceptibility and MIC Determination

The human pathogen *C. albicans* SC 5314 used in this study was from the Institute of Microbiology, Chinese Academy of Science. It was stored in 25% glycerol at −80 °C. Antifungal susceptibility testing was implemented as described previously [39] in flat bottom, 96-well microtiter plates, using a broth micro-dilution protocol modified from the CLSI M-27A3 methods [40]. Instead of RPMI 1640 medium, YNB medium was used for the antifungal assay since it can avoid the influence of branched-chain amino acids [18,19]. The concentration of fungal cells in the testing media was 3 × 10^5^ CFU/mL. MICs were determined as the minimum inhibitory concentrations of target compounds that inhibit fungal growth relative to the corresponding drug-free growth control after 24 h incubation. The experiments were performed twice in triplicate. Fluconazole (FCZ) purchased from Sigma was used as a positive control drug.

### 3.6. Single Crystal X-ray Diffraction

A colorless crystal with dimensions of 0.23 × 0.17 × 0.2 mm of **9j** was obtained by self-evaporation in acetone solution. It was mounted on a glass fiber for X-ray diffraction analysis, and the data was collected at 113.15 K on a Rigaku Saturn 70 CCD diffractometer with graphite monochromated MoKα radiation (λ = 0.71073 Å), *θ*_max_ = 32.812°. The chemical structures were solved using the “Direct methods” techniques of the Bruker software package SHELXTL [41]. All calculations were refined anisotropically.

### 3.7. Density Functional Theory (DFT) Calculation

Except **9j** itself, the molecular structures of **9i**, **9q**, and **9u** were constructed based on the crystal structure of **9j** using GaussView 5.0 [41]. The DFT calculations were performed using Gaussian 09 [42]. The geometries were optimized by the SCF (Ground State) method using the B3LYP function (em=GD3BJ) with a basis set of 6-31G (d) to describe their molecular properties. All computations were conducted for the ground states of these molecules as singlet states. All of the convergent precisions used were the system’s default values. HOMO and LUMO maps were visualized by GaussView 5.0.

## 4. Conclusions

The prevalence of the diseases caused by MRSA has made it a serious challenge for medicinal treatments, considering the fact that a very limited number of antibiotics are effective against such superbug infections. The invention of novel antibacterial agents is, therefore, in great demand. However, the discovery of the bioactive compounds with different chemical skeletons is becoming more and more difficult. One routine idea for drug discovery is to identify active compounds from the microbial metabolites. However, this relies highly on the research of natural products, and these resources are nearly exhausted. Another approach which brings “classic” compounds to “new” bioassay platforms to search for unexpected biological activity, turns out to be a preferable solution since there exists a considerable drug-likeness opportunity. The sulfonylureas are a very “old” family of compounds that have been used as either herbicides or antidiabetic drugs for a long period, indicating that this class of compounds have great potential for the development of new antibacterial agents if such activity can be identified from a new biological assay. Herein, we present for the very first time that some of novel sulfonylurea compounds are potent inhibitors of Gram-positive bacteria, especially for clinically isolated MRSA strains, and these results have provided meaningful clues for further development of novel antibacterial agents to overcome drug resistance of pathogen bacteria. We must admit that this is just a new starting point for medicinal research, and it is obvious that a full evaluation must be conducted for each biologically active compound to determine whether it is worth paying further attention to this family of compounds as anti-MRSA agents. For this purpose, drug properties of absorption, distribution, metabolism, and excretion (ADME) should be studied using an animal model, as well as the toxicity behavior. In the next stage, it is necessary to carry out an extensive structural optimization to discover new compounds with ten-fold or even more potent activity, and to evaluate the cytotoxicity of the synthesized compounds. On the other hand, the molecular target for the anti-MRSA activity needs to be explored and elucidated to understand the structure–activity relationships of this family of antibacterial agents. Overall, this research has offered some fresh insights for the identification and discovery of anti-MRSA agents with chemical structures different from the available antibacterial drugs.

## Data Availability

Data is contained within the article or Appendix A.

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
