# Peer review of "Novel Sulfonylurea Derivatives as Potential Antimicrobial Agents: Chemical Synthesis, Biological Evaluation, and Computational Study"

_antibiotics, 2023, doi:10.3390/antibiotics12020323_

Round 1

Reviewer 1 Report

The manuscript of Meng et al entitled “Novel Sulfonylurea Derivatives as Potential Antibacterial Agents: Chemical Synthesis, Biological Evaluation, and Computational Study” describes the synthesis and biological evaluation of novel sulfonylurea derivatives. The manuscript represents a good scientific contribution by presenting a new class of compounds with prominent biological activity in addition to presenting a very robust chemical synthesis. However, I believe that the authors could evaluate some points as described below.

1. The figure 1 should be introduced in the text in “Especially, compound A (Figure 1) exhibited…”

2.  In scheme 1, please, include the temperature at which the reactions were carried out.

3. Concern the sentence “As illustrated in Scheme 1, the methanol generated in the reaction mixture should be removed continuously to accelerate the reaction rate for the preparation of intermediate 2.”: the generation of methanol in the reaction may not be clear to all readers, because the authors have not shown it and no indication of temperature was afforded in this reaction. Furthermore, this information is not relevant to this discussion. However, this is very important for the procedure, thus I recommend moving it to “3. Materials and Methods”

4. In “it was allowed to cool to room temperature and sufficient n-hexane was added under stirring conditions”, please, inform the exact amount of n-hexane that was used. Similar to “Sufficient water was added to the reaction mixture under stirring conditions…” Please, inform the amount of water added.

5. In “The precipitate was filtrated and dried in vacuo to give 4a-i.” please, inform if a purification step was not necessary.

6. In the general description of “Synthesis of Intermediates 4j-l [35]”, seems that the authors describe the conversion of 4b (Ar=4-OCH3-Ph) to 4j (2-furanyl), but it is not correspondent to scheme 1. In this scheme, the representation of the conversion of 4a-i to 4a-i sounds not appropriate. 9 Compounds (4a-i) were converted into three (4a-i). The authors commented something about it, but the understanding of a scheme should be independent of the text.

7. In scheme 1, seems to me that compounds 5a-d are obtained as a by-side product. The same reaction produces 4m-p and 5a-d. Only in the procedure that I realize that 5a-d are obtained in a separate procedure. The authors should improve the scheme to make this clear. In the procedure of 5a-d, is the adjustment of pH value of the solution of the residue in water that produce 5a-d instead of 4m-p? This part is not clear to me. I suggest the authors explain in detail the procedure to obtain 5a-d and separate them in scheme 1.

8. In the representation of substituents in molecule 9, in scheme 1, the authors could make a table similar to the first 4 columns of table 2. If the authors break this table into two columns, it will occupy the same space as the representation of substituents in molecule 9 in Scheme 1.

9. In “To the suspended mixture, aqueous hydrochloric acid (1 M) solution was added to quench the reaction.”, please, inform the amount of the aqueous hydrochloric acid (1 M) solution used.

 10. In several sentences the authors say something like “the reaction was monitored by TLC till it completed”. The authors need to include the time of the reaction for each case. If the time of reaction was similar for the reactions, the authors may include a range of time. This is very important information. I do not know if this reaction is carried out in 1 hour or 1 day.

Author Response

Comments and Suggestions for Authors

The manuscript of Meng et al entitled “Novel Sulfonylurea Derivatives as Potential Antibacterial Agents: Chemical Synthesis, Biological Evaluation, and Computational Study” describes the synthesis and biological evaluation of novel sulfonylurea derivatives. The manuscript represents a good scientific contribution by presenting a new class of compounds with prominent biological activity in addition to presenting a very robust chemical synthesis. However, I believe that the authors could evaluate some points as described below.

Answer: Thanks for your encouraging comments on our research.

  1. The figure 1 should be introduced in the text in “Especially, compound A (Figure 1) exhibited…”

Answer: We have made a change based on your comment.

  1. In scheme 1, please, include the temperature at which the reactions were carried out.

Answer: Thanks for the kind reminder. In the revised version all the temperatures were given, in both Scheme 1 and the materials and methods.

  1. Concern the sentence “As illustrated in Scheme 1, the methanol generated in the reaction mixture should be removed continuously to accelerate the reaction rate for the preparation of intermediate 2.”: the generation of methanol in the reaction may not be clear to all readers, because the authors have not shown it and no indication of temperature was afforded in this reaction. Furthermore, this information is not relevant to this discussion. However, this is very important for the procedure, thus I recommend moving it to “3. Materials and Methods”

Answer: Yes, we agree that it is better to move this sentence to the materials parts. Now it is in the section of “3.2.1. Synthesis of Intermediates 2 [32]”

  1. In “it was allowed to cool to room temperature and sufficient n-hexane was added under stirring conditions”, please, inform the exact amount of n-hexane that was used. Similar to “Sufficient water was added to the reaction mixture under stirring conditions…” Please, inform the amount of water added.

Answer: We appreciate your careful reading of the manuscript. In the revised paper, the exact amount was provided for all the liquids like this. 

  1. In “The precipitate was filtrated and dried in vacuo to give 4a-i.” please, inform if a purification step was not necessary.

Answer: No purification is needed here. We have added “without further purification” after this sentence.

  1. In the general description of “Synthesis of Intermediates 4j-l [35]”, seems that the authors describe the conversion of 4b (Ar=4-OCH3-Ph) to 4j (2-furanyl), but it is not correspondent to scheme 1. In this scheme, the representation of the conversion of 4a-i to 4a-i sounds not appropriate. 9 Compounds (4a-i) were converted into three (4a-i). The authors commented something about it, but the understanding of a scheme should be independent of the text.

Answer: Thanks for your careful checking. In fact 4g-I were converted to 4g-l. We have made corresponding revision for Scheme 1.

  1. In scheme 1, seems to me that compounds 5a-d are obtained as a by-side product. The same reaction produces 4m-p and 5a-d. Only in the procedure that I realize that 5a-d are obtained in a separate procedure. The authors should improve the scheme to make this clear. In the procedure of 5a-d, is the adjustment of pH value of the solution of the residue in water that produce 5a-d instead of 4m-p? This part is not clear to me. I suggest the authors explain in detail the procedure to obtain 5a-d and separate them in scheme 1.

Answer: Thanks for your valuable comments and suggestions. Compounds 4m-p and 5a-d could be obtained via similar synthetic procedure and reaction mechanism, but the compounds 5a-d are not the by-products. The starting materials of compounds 4m-p are ethyl acetate and compound 1, and the starting materials of compounds 5a-d are diethyl carbonate and compound 1. We also made some change in Scheme 1 to facilitate understanding for readers.

  1. In the representation of substituents in molecule 9, in scheme 1, the authors could make a table similar to the first 4 columns of table 2. If the authors break this table into two columns, it will occupy the same space as the representation of substituents in molecule 9 in Scheme 1.

Answer: In fact, it is a bit redundant to represent the substituents in molecule 9, since these have been illustrated in Table 1. Therefore, we removed this part in Scheme 1.

  1. In “To the suspended mixture, aqueous hydrochloric acid (1 M) solution was added to quench the reaction.”, please, inform the amount of the aqueous hydrochloric acid (1 M) solution used.

Answer: similar to the answer to comment 4, the amount was given.

  1. In several sentences the authors say something like “the reaction was monitored by TLC till it completed”. The authors need to include the time of the reaction for each case. If the time of reaction was similar for the reactions, the authors may include a range of time. This is very important information. I do not know if this reaction is carried out in 1 hour or 1 day.

Answer: Thanks for pointing this out. In the revised paper the reaction time for each case was given. Please check these in the synthesis of intermediates and target compounds.

Reviewer 2 Report

The article describes the chemical development, antibacterial activity and computational study of some new sulfonylureea compounds.

General remarks

- the research hypothesis and the chemistry part are interesting

- the structural analysis of the compounds is correctly performed

- the antimicrobial activity determination and the discutions of the results are correct

- in general, the research is of good quality

Recommendations and corrections:

- to change in the title antibacterial with antimicrobial. Candida sp it is a fungus.

- to give up (Chaoyang) after MRSA, with the exception of the Material and method section. Or, to write (Chaoyang clinical isolates) in the whole article

- In the introduction, on page 2, what does weeding activity mean?

- in the introduction, page 2, it can be mentioned that there are many licensed antidiabetic sulphonylureas, more important than acetohexamide. For example glipizide and glyburide.

- page 2, line 4, MRSA infections

- the generic name arylacetophenone for compounds 1 is not correct. I propose aryl-methyl-ketones. For arylacetophenone name, the acetophenone should be substituted with aryl-residue. It is not the case here.

- page 4, last line, to complete after azeotrope with mixture

- page 11, at synthesis of 9a-u, why NaHCO3 solution and brine?

Author Response

The article describes the chemical development, antibacterial activity and computational study of some new sulfonylureea compounds.

General remarks

- the research hypothesis and the chemistry part are interesting

- the structural analysis of the compounds is correctly performed

- the antimicrobial activity determination and the discutions of the results are correct

- in general, the research is of good quality

Answer: We really appreciate your positive comments on our manuscript.

Recommendations and corrections:

- to change in the title antibacterial with antimicrobialCandida sp it is a fungus.

Answer: We have changed the title accordingly.

- to give up (Chaoyang) after MRSA, with the exception of the Material and method section. Or, to write (Chaoyang clinical isolates) in the whole article’

Answer: Thanks for the suggestion, we have used (Chaoyang clinical isolates) for the revision in the whole article.

- In the introduction, on page 2, what does weeding activity mean?

Answer: weeding activity means herbicidal activity. In the revised paper we used “herbicidal activity” instead.

- in the introduction, page 2, it can be mentioned that there are many licensed antidiabetic sulphonylureas, more important than acetohexamide. For example glipizide and glyburide.

Answer: This is a good suggestion. We now used glipizide as an example. Therefore, Figure 1 and ref 16 were updated.

- page 2, line 4, MRSA infections

Answer: Thanks. We have corrected this in the paper.

- the generic name arylacetophenone for compounds 1 is not correct. I propose aryl-methyl-ketones. For arylacetophenone name, the acetophenone should be substituted with aryl-residue. It is not the case here.

Answer: Thanks for pointing this out. We have used aryl-methyl-ketones instead in the revision.

- page 4, last line, to complete after azeotrope with mixture

Answer: we added mixture after azeotrope in the paper (twice).

- page 11, at synthesis of 9a-u, why NaHCO3 solution and brine?

Answer: We have checked our original lab records. In fact, brine is not essential for the synthesis. Therefore, we deleted “and brine” throughout the manuscript.
